# Bisphenol A and Bisphenol S in Hemodialyzers

**DOI:** 10.3390/toxins15070465

**Published:** 2023-07-20

**Authors:** Zahin Haq, Xin Wang, Qiuqiong Cheng, Gabriela F. Dias, Christoph Moore, Dorothea Piecha, Peter Kotanko, Chih-Hu Ho, Nadja Grobe

**Affiliations:** 1Renal Research Institute, New York, NY 10065, USA; 2Fresenius Medical Care North America, Waltham, MA 02451, USA; 3Fresenius Medical Care (Germany), 61352 Bad Homburg, Germany; 4Icahn School of Medicine at Mount Sinai, New York, NY 10029, USA; 5Fresenius Medical Care North America, Ogden, UT 84404, USA

**Keywords:** hemodialysis, toxicology, LC–MS, extractable, leachable, polysulfone, polyethersulfone, uremic toxin

## Abstract

Bisphenol A (BPA)-based materials are used in the manufacturing of hemodialyzers, including their polycarbonate (PC) housings and polysulfone (PS) membranes. As concerns for BPA’s adverse health effects rise, the regulation on BPA exposure is becoming more rigorous. Therefore, BPA alternatives, such as Bisphenol S (BPS), are increasingly used. It is important to understand the patient risk of BPA and BPS exposure through dialyzer use during hemodialysis. Here, we report the bisphenol levels in extractables and leachables obtained from eight dialyzers currently on the market, including high-flux and medium cut-off membranes. A targeted liquid chromatography–mass spectrometry strategy utilizing stable isotope-labeled internal standards provided reliable data for quantitation with the standard addition method. BPA ranging from 0.43 to 32.82 µg/device and BPS ranging from 0.02 to 2.51 µg/device were detected in dialyzers made with BPA- and BPS-containing materials, except for the novel FX CorAL 120 dialyzer. BPA and BPS were also not detected in bloodline controls and cellulose-based membranes. Based on the currently established tolerable intake (6 µg/kg/day), the resulting margin of safety indicates that adverse effects are unlikely to occur in hemodialysis patients exposed to BPA and BPS quantified herein. With increasing availability of new data and information about the toxicity of BPA and BPS, the patient safety limits of BPA and BPS in those dialyzers may need a re-evaluation in the future.

## 1. Introduction

Bisphenol A (BPA, 2,2-bis-(4-hydroxyphenyl) propane) is a widely utilized industrial chemical and is commonly applied in the manufacturing of polycarbonate (PC) plastics. PC is used in a variety of industries and applications, but its use in food and beverage containers as well as medical products has raised attention due to the potential migration of BPA into human circulation [1]. Over the last decade, a number of studies have characterized BPA as an endocrine-disrupting agent. Toxicity studies have indicated that the kidney and liver are relevant target organs for BPA toxicity [2]. Further possible effects have been described on the reproductive, nervous, immune, metabolic, and cardiovascular systems [3], as well as on the development of cancer [4] and albuminuria [5,6]. These health concerns have led to the development of several “BPA-free” consumer and health products. However, most of the chemical replacements for BPA also contain bisphenols. Such substitutes, including Bisphenol S (BPS) and other similar compounds, have not been intensively studied and are considered nonideal because of their possible similar physiological effects in organisms. PC, alongside BPA, is commonly used in high-volume medical device disposables, such as catheters, tubing, and hemodialyzer housing. Furthermore, polysulfone (PS) and polyethersulfone (PES), widely used in hemodialysis membranes, contain BPA and BPS, respectively [7]. BPA use is permitted by regulatory bodies for the general population due to its almost complete renal clearance. Yet, this implies that serum BPA levels increase with decreasing renal function. In fact, studies have shown that the levels of BPA and BPS are highest in individuals on hemodialysis compared to matched healthy controls [7,8,9]. The structural resemblance between bisphenols and established phenolic uremic toxins, such as p-cresol and its derivatives, as well as similar metabolism and disposal compared with uremic toxins, raise concerns about aggravated harmful effects in anuric dialysis patients [10] (Figure 1).

Therefore, the safety of bisphenol exposure in the general population cannot be extrapolated to a dialysis population. Consequently, leading dialyzer manufacturers mitigate potential leaching through several strategies, including using BPA/S-free housing materials, such as polypropylene (PP), and membrane materials, such as polyvinylpyrrolidone (PVP) and cellulose triacetate.

Given the already elevated inflammatory status (reviewed in [11]) in patients with kidney failure, the broad application of BPA in medicinal products, as well as the repetitive nature of hemodialysis therapy, it is important to determine the amounts of BPA and BPS that are leached from dialyzers. Work has been performed previously in assessing the safety use of dialyzers manufactured by Fresenius regarding BPA exposure [12]. In the current project, eight commercial dialyzers manufactured by Fresenius and other companies were included to investigate the extraction and leaching properties of BPA and BPS under simulated and extreme conditions using targeted liquid chromatography–mass spectrometry (LC–MS).

## 2. Results

Eight different dialyzers were tested in this study (Table 1). These dialyzers are manufactured with different materials used in the membrane and housing. PS and PC are BPA-based materials, and PES is a BPS-based material, while PVP, PP, and cellulose or asymmetric triacetate are neither BPA- nor BPS-based materials. Aside from the use of distinct materials, they also undergo different sterilization processes. The extractable and leachable amounts of BPA and BPS of each dialyzer were quantified using the standard addition method (SAM), and the results are listed in Table 1. BPA (0.43–32.82 µg/device) and BPS (0.02–2.51 µg/device) were detected in extractables from dialyzers and housings made with BPA- and BPS-containing materials, respectively. In leachables, BPA was only detected in one dialyzer made of BPA-containing membrane and housing material. BPS (0.08–1.44 µg/device) was detected in leachables of dialyzers made with BPS-based materials. Fresenius Medical Care’s FX CorAL 120 dialyzer is manufactured using PS, a BPA-containing material. However, bisphenols were not detected in the extractables and leachables of the FX CorAL 120. Representative chromatograms for BPA and BPS and their corresponding stable isotope-labeled internal standards, [^13^C_12_]-BPA and [^13^C_12_]-BPS, are shown in Figure 2. For all dialyzers, the limit of quantitation for the bloodline control was between 0.12–6 ppb for BPA and 0.08–0.24 ppb for BPS. BPA/BPS were not detected in the bloodline controls and cellulose-based membranes.

## 3. Discussion

In this study, the extraction and leaching profiles of BPA and BPS from dialyzers were evaluated. The extracts from Sureflux and Solacea dialyzers are BPA- and BPS-free because both dialyzers have cellulose-based membranes and BPA-free PP housing. In extracts from all tested dialyzers that contain either PS membranes or PC housings, BPA and/or BPS were detected. The FX CorAL dialyzer 120, a member of the recently FDA-cleared CorAL dialyzer family, uses BPA-containing PS as the membrane material and PP housing. BPA was not detected in FX CorAL extracts, which could be attributed to adding PVP to the surface of the fiber on the blood side as well as using membrane stabilizers and steam sterilization. These extra measures may have minimized BPA leaching out from the membrane.

Although there are persistent concerns regarding potential health risk for BPA exposure, particularly reproductive and or developmental effects, the predominant view is that BPA, at the level used in consumer products, is associated with minimal health risks for the general population [13,14]. For kidney disease patients on hemodialysis, repeated exposure to BPA- and BPS-leaching catheters, tubing, filters, and housing is a major concern. BPA and BPS bind to serum albumin up to 70% and removal by hemodialysis remains limited [15,16]. Like other protein-bound uremic toxins, enhanced removal using high-flux and medium cut-off dialyzers in combination with hemodiafiltration reduce bisphenol levels in dialysis patients [17]. Other strategies can be implemented to reduce the risks of bisphenol levels in these patients [18,19].

For the quantitative risk assessment, a human equivalent dose of 609 µg/kg/day is typically used as the reference point. This was established by The European Food Safety Authority [13] based on kidney effects observed from several studies including GLP-compliant multigeneration reproductive toxicity studies that followed regulatory testing guidelines where the lowest No Observed Adverse Effect Level (NOAEL) was set at 5 mg/kg/day [20,21]. The uncertainty regarding BPA’s effects on mammary glands and reproductive, neurobehavioral, immune, and metabolic systems has been continuously recognized by regulatory authorities and expert opinion panels.

BPS-containing materials, such as PES, are used as an alternative to PS for dialyzer membrane production. Based on structural similarity, BPS is expected to have a similar toxicological profile to BPA. The estrogenic activity of BPS is less than that of BPA, but both are of comparable potency [22,23,24]. Research studies on the health impacts of BPS have not been reported frequently because of the increased use of PES. At this stage, compared to BPA, limited information is available on BPS. Thus, a systemic review and a comprehensive risk assessment on BPS for its exposure are needed [2].

Regarding the toxicological concern of the safe use of medical devices, the Margin of Safety (*MOS*) serves as a measure of determination of acceptable health risk. If the *MOS* is >1, then it is unlikely for adverse effects to occur in the exposed patients. In general, the *MOS* is calculated using the following equation:$$MOS=\frac{Tolerable\;Intake\left(\mu g/kg/day\right)}{Patient\;Exposure\left(\mu g/kg/day\right)}$$

The *Tolerable Intake* (*TI*) of BPA in medical devices derived by the Scientific Committee on Emerging and Newly Identified Health Risks is 6 μg/kg/day [2]. Given BPS and BPA’s structural similarity and shared toxicity profiles, BPA’s *TI* can be used to calculate the *MOS* of BPS. With the extraction amount reported here, the daily exposure of BPA and BPS to patients through dialyzer use can be estimated with the following assumptions: (1) one dialyzer per hemodialysis session; (2) three sessions per week for long-term patient treatment; (3) a patient body weight of 60 kg. Using the *TI* of BPA established from the available toxicity data and assuming a *TI* for BPS similar to BPA, an *MOS* can be calculated and serves as a measure of determination of acceptable health risk. The *MOS*’s of all the dialyzers investigated herein are many-fold higher than 1, indicating that it is unlikely for adverse effects to occur in exposed hemodialysis patients. Specific calculation of the *MOS* for BPA and BPS from the dialyzers investigated in the current project was performed but not published (can be shared upon request). However, the safety of dialyzers made with BPA- and/or BPS-containing materials warrants re-evaluation whenever new data regarding toxicity of BPA and/or BPS emerge. Nevertheless, as a general precaution, nephrologists should aim to apply techniques that result in a reduced exposure to bisphenols.

## 4. Conclusions

In our study, we did not detect any BPA or BPS in the hemodialyzers that use non-BPA- or non-BPS-containing materials. Because the BPA- or BPS-containing materials have been widely used in dialyzer manufacturing industries, there is a higher likelihood that dialysis exposes kidney disease patients to BPA and BPS compounds. Compared with other uremic toxins, it is desirable to avoid additional loads of potentially harmful BPA and BPS through hemodialyzers and housing made of them. The leaching of BPA or BPS depends on the manufacturing processes and sterilization methods. The extractables and simulated-use leachable results for the Fresenius’ FX CorAL 120 and cellulose-based membranes demonstrate that BPA or BPS were not detected. The risk assessment using currently available data indicates that it is unlikely for adverse effects to occur in hemodialysis patients exposed to the bisphenol levels reported herein.

## 5. Materials and Methods

### 5.1. Sample Preparation

Samples for BPA and BPS analysis were extracted from the dialyzers in extractable (exaggerated) and leachable (simulated-use) conditions in triplicate. Prior to extraction, the dialyzer was primed in a Fresenius 2008T dialysis machine according to the device’s instructions for use. For extractables, the dialysate side solution was replaced with 95% ethanol (EtOH), with volume recorded, and sealed. With a benchtop pump, the priming solution in the blood side was replaced with 95% EtOH using the exact fill volume of the bloodlines and dialyzer. The bloodlines were clamped during the static incubation at 37 °C for 72 h. After 72 h, the extraction solutions from the dialysate side and blood side were collected separately for BPA and BPS analyses. As much solution as possible was collected from the dialysate side before the blood side. The extraction solution from bloodline was not included in the collection.

For leachables, the dialysate side solution was replaced with 17.2% EtOH, with volume recorded, and sealed. The blood side of the dialyzer was connected to a reservoir containing 1 L of 17.2% EtOH through bloodline and blood pump. The dialyzer was filled with 17.2% EtOH and underwent a recirculation at a blood flow rate of 300 mL/min at 37 °C for 24 h. The schematic of the leachable condition is shown in Figure 3.

After 24 h, the recirculation was stopped, and the solution was collected for further BPA and BPS analyses. The solution from the dialysate side was not collected, as an equilibrium should have been reached between the dialysate side and blood side in the dialyzer during the recirculation. A bloodline control was run in parallel for each type of dialyzer under the simulated-use extraction, where no dialyzer was included in the circuit under the same setting of recirculation system. The Fresenius Combi bloodline was used. It was made from polyvinyl chloride, and the fill volume of the bloodline was 145 mL.

All extractables and leachables were shipped to the RRI laboratory for analysis.

### 5.2. Liquid Chromatography–Mass Spectrometry (LC–MS) Analysis

The RRI laboratory staff were blinded to the dialyzer names and manufacturers. BPA and BPS levels in each batch were assessed using an Agilent 1290 High-Performance Liquid Chromatograph connected to an Agilent Ultivo Triple Quadrupole instrument. The mass spectrometer operated in negative ionization mode using a Jet Stream electrospray ionization source. The source parameters were optimized to the following values: Gas Temperature: 150 °C, Gas Flow: 12 L/min, Nebulizer Pressure: 45 psi, Sheath Gas Temperature: 250 °C, Sheath Gas Flow: 10 L/min, Capillary Voltage: 5500 V, Nozzle Voltage: 2000 V. A Waters VanGuard BEH C18 precolumn (1.7 µm particle size, 2.1 mm × 5 mm length) was connected to a Waters Acquity UPLC BEH C18 column (1.7 µm particle size, 2.1 × 50 mm length), and a gradient mobile phase consisting of 0.1% acetic acid and acetonitrile was used for chromatographic separation. The initial gradient condition of 40% acetonitrile linearly increased to 98% in 3 min, where it was held for 30 s before returning to initial conditions and being held for another 3 min. An injection volume of 10 µL and a flow rate of 0.6 mL/min were used.

With this approach, selected ions (precursor ions) were broken down into fragments (product ions) using collision-induced dissociation. For example, the precursor ion for bisphenol A is *m/z* 227.1. After applying collision-induced dissociation, two product ions with *m/z* 212.2 and *m/z* 133.1 were detected in high abundance. The collision-induced dissociation of precursor ion *m/z* 227.1 into the product ions *m/z* 212.2 and *m/z* 133.1 was optimized, and the final collision energy for each product ion is shown in Table 2.

### 5.3. Quantification

BPA and BPS concentrations were quantified using Agilent MassHunter QQQ Quantitative Analysis. A SAM was set up for each unspiked sample and consisted of spiked concentrations of unlabeled and [^13^C_12_]-BPA/BPS. Advantages of SAM over traditional quantification methods are (1) extractables and leachables are highly complex samples with lots of background noise and already contain the targets of interest, BPA and BPS; (2) a substitute or blank matrix representing extractables and leachables without the targets BPA and BPS is not available; (3) the recovery rates are high due to similar interferences caused by the complexity of the matrix. A detailed overview of SAM is given elsewhere [25]. Briefly, the SAM calibration curve consisted of a series of five concentrations ranging from 0.24 ppb to 150 ppb of BPA (certified reference material, TraceCERT, 42088-100MG, Sigma-Aldrich, St. Louis, MI, USA) and BPS standards (analytical standard, 43034-100MG, Sigma-Aldrich, St. Louis, USA) added to the unspiked extractable or leachable. [^13^C_12_]-BPA internal standard (CLM-4325-1.2, Cambridge Isotope Laboratories) and [^13^C_12_]-BPS internal standard (CLM-9319-1.2, Cambridge Isotope Laboratories) were mixed into all samples. The SAM calibration curves were created in duplicate for each leachable and extractable. Retention times and product ion transition values (Qual/Quan ratio) were used for quality control. Samples were quantifiable if the retention time was within 3.1% and Qual/Quan ratio of labeled BPA or BPS and unlabeled BPA or BPS was 100% ± 20%. SAM calibration curves were created using the ratio of unlabeled to labeled standard. A prerequisite was that accuracy and recovery for each point on the SAM calibration curve were within 100% ± 20%.

## Figures and Tables

**Figure 1 toxins-15-00465-f001:**
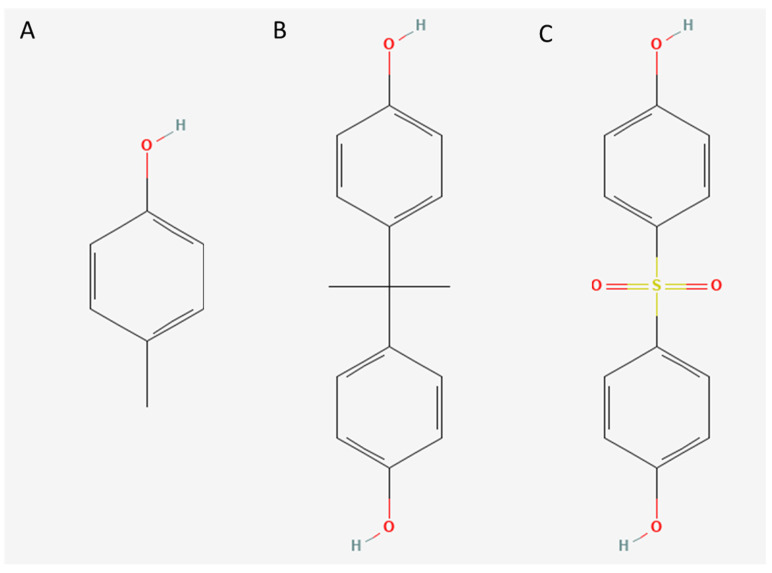
Structural resemblance between established uremic toxin p-cresol and bisphenols. (**A**) p-cresol. (**B**) Bisphenol A. (**C**) Bisphenol S. All structures were obtained from https://pubchem.ncbi.nlm.nih.gov/ (accessed on 24 May 2023).

**Figure 2 toxins-15-00465-f002:**
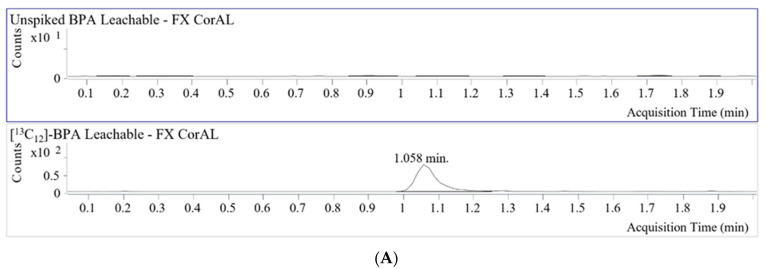
Example BPA/BPS chromatograms for FX CorAL 120 leachables. (**A**) BPA chromatogram of the unspiked FX CorAL leachable along with its corresponding [^13^C_12_]-BPA chromatogram. BPA was not detected. (**B**) Chromatogram of 6 ppb BPA spiked into FX CorAL leachable along with its corresponding [^13^C_12_]-BPA chromatogram. Spiked unlabeled amount of BPA (6 ppb) was based on the limit of detection for the unspiked FX CorAL leachable. (**C**) BPS chromatogram of the unspiked FX CorAL leachable along with its corresponding [^13^C_12_]-BPS chromatogram. BPS was not detected. (**D**) Chromatogram of 0.24 ppb BPS spiked into FX CorAL leachable along with its corresponding [^13^C_12_]-BPS chromatogram. Spiked unlabeled amount of BPS (0.24 ppb) was based on the limit of detection for the unspiked FX CorAL leachable.

**Figure 3 toxins-15-00465-f003:**
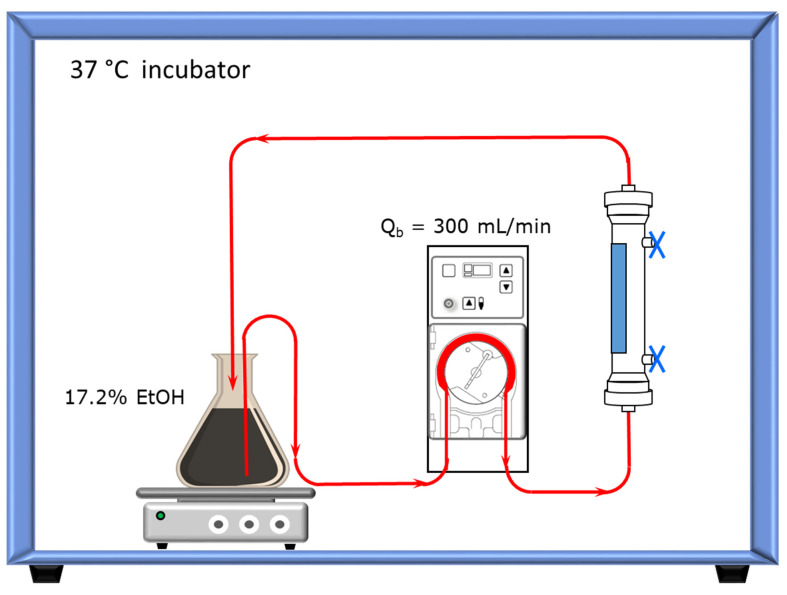
Schematic of leachable condition. The blood side of each dialyzer was connected to a reservoir containing 1 L of 17.2% EtOH, which was recirculated through bloodline and blood pump at 300 mL/min and 37 °C for 24 h.

**Table 1 toxins-15-00465-t001:** BPA and BPS extraction masses (µg/device) in extractables (Es) and leachables (Ls) of tested dialyzers made with different materials and manufacturing processes.

Dialyzer Type	Manufacturer	Membrane Materials	Housing Materials	Sterilization	BPA (µg/Device)		BPS (µg/Device)	
					Es	Ls	Es	Ls
FX CorAL 120	Fresenius	PS/PVP	PP	In-line steam	ND	ND	ND	ND
Theranova 500	Baxter	PAES/PVP	PC	Steam	0.73	ND	0.02	0.08
Elisio 21H	Nipro	PES	PP	Gamma ray	ND	ND	1.37	0.96
Sureflux 21UX	Nipro	Cellulose Triacetate	PP	Gamma ray	ND	ND	ND	ND
Solacea 21H	Nipro	Asymmetric triacetate	PP	Gamma ray	ND	ND	ND	ND
Xevonta Hi23	B. Braun	PS/PVP	PC	Gamma ray	32.82	2.48	1.99	1.44
Revaclear 400	Baxter	PAES/PVP	PC	Steam	0.43	ND	0.27	0.15
Bain B-20HF	Bain	PES	PP	Radiation	ND	ND	2.51	0.62

PS—polysulfone, PVP—polyvinylpyrrolidone, PP—polypropylene, ND—not detected, PAES—polyarylethersulfone, PC—polycarbonate, PES—polyethersulfone. PS and PC are BPA-based materials, and PES is a BPS-based material, while PVP, PP, and cellulose or asymmetric triacetate are neither BPA- nor BPS-based materials. Different sterilization processes are indicated. 95% ethanol is extractable; 17.2% ethanol is leachable. All samples were analyzed in duplicate for three technical replicates. For the 95% EtOH extracts, the BPA and BPS amounts in the extracts of the blood side and dialysate side were combined and reported as the total amounts in each sample for each dialyzer. For calculating the extraction mass in the extractables, the extraction volume and density of 0.8 g/mL for 95% EtOH was used. For leachables, no density adjustment was used as the density of 17.2% EtOH is close to 1 g/mL.

**Table 2 toxins-15-00465-t002:** Optimized parameters for targeted LC–MS analysis of unlabeled BPA/BPS and [^13^C_12_]-labeled BPA/BPS. Two product ions per precursor ion were monitored; one product ion is the quantifier (Quan), and the other product ion is the qualifier (Qual). The ratio between the area under the curve for the signal of Qual to the area under the curve for the signal of Quan is an important criterion for identifying compounds.

Compound Name	Precursor (*m*/*z*)	Product (*m*/*z*)	Dwell Time (ms)	Collision Energy (V)	Retention Time (min)
Bisphenol A	227.1	212.2 (Quan)	120	21	1.03
Bisphenol A	227.1	133.1 (Qual)	120	29	1.03
Bisphenol S	249	108 (Quan)	120	33	0.51
Bisphenol S	249	92.1 (Qual)	120	37	0.51
[^13^C_12_]-Bisphenol A	239.2	224.1 (Quan)	120	21	1.03
[^13^C_12_]-Bisphenol A	239.2	139 (Qual)	120	33	1.03
[^13^C_12_]-Bisphenol S	261.1	114 (Quan)	120	33	0.50
[^13^C_12_]-Bisphenol S	261.1	98 (Qual)	120	45	0.50

## Data Availability

The data presented in this study are available on request from the corresponding author.

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
