# Peer review of "Bisphenol A and Bisphenol S in Hemodialyzers"

_toxins, 2023, doi:10.3390/toxins15070465_

Round 1

Reviewer 1 Report

This study investigated the exposure of dialysis patients to two substances released during dialysis from the housing and membranes in hemodialyzers, Bisphenol A (BPA) and Bisphenol S (NPS). BPA and BPS values were extracted from samples of 8 different dialyzers from various manufacturers currently in use, six with polysulfone or derivatives (PS) membranes and two with cellulosic membranes, in extractable and leachable conditions. All polysulfone (PS) or PS derivatives dialyzers released small amounts of BPA and/or BPS. However, the released quantities of bisphenols were lower than the levels currently considered harmful to health for which the authors conclude that the "risk assessment using currently available data indicates that is unilkely for adverse effects to occur in exposed hemodialysis patients". The study was well conducted and well written, so it can be published in its current form. I would just like to point out that, as the authors establish, the text states that bishenols can be contained both in the housing (Polypropylene, PP or Polycarbonate, PC) and in the membranes. The housings of the three dialyzers that did not release bisphenols were in Polypropylene, exactly like the housings of 2 out of the three dialyzers that released bisphenols. Is it possible to deduce that the contribution of the PP or PC contained in the housings is irrelevant to the low level of bisphenols released?

Reviewer 2 Report

The authors present data on an analysis of potential bisphenal B (BPA) and bisphenol S (BPS) exposure from a number of commonly used hemodialyzers. They show that BPA and BPS are not detectable in fluids exposed to a specific PS-containing dialyzer from Fresenius and cellulose triacetate and asymmetric triacetate-based dialyzers, while other investigated dialyzers have higher extractable and leachable amounts of BPA and BPS. They conclude that these amounts are likely to be well under tolerable exposure limits. This is a relevant, current subject that has garnered little attention thus far in the nephrology literature.

The manuscript is well written. I do have a number of questions.

Major:

11    Due to the relatively prominent presentation of the FX CorAL 120 hemodialyzer (first in Table 1, all of Figure 2), which was only recently introduced to the market, is being actively marketed, and has a much smaller market share than Fresenius’ other hemodialyzers (amongst which, the CorDiax dialyzers), the manuscript reads a bit like an advertisement. Is there a conflict of interest by the authors? Why did the authors choose not to analyze Fresenius’ other dialyzers?

22     Page 7, line 200-201. For which of the solutions (dialysate side, blood side) are data presented here?

Minor:

11    Page 1, line 38. Please leave out “pathological”

2.      Page 2, line 63. Preferred (KDIGO) nomenclature is kidney failure (instead of end-stage kidney disease)

32.    Page 2, line 63. Does inflammation worsen BPA toxicity?

43.      Table 1. Amembris is a trade name, please leave out.

54.      Page 7, line 198. How was the EtOH solution brought into the blood side from the dialysate side?

65.      Page 7, line 204. Why was 17.2% EtOH chosen?

76.      Page 7, line 208. Recirculation was done for 24 hours. A typical dialysis session has a duration of 24 hours and in most cases the blood EtOH concentration is lower than 17.2%. How do the authors expect this to affect projections of actual exposure

87.      Page 8, 216. Please show data for the bloodline control, and please show details of the material used (volume, type).

98.      Page 5, lines 120-124 are a repetition of the introduction and may be (re)moved there

19.  Page 6. Can the authors speculate about 1) the additional/synergetic effects of BPA and BPS exposure; 2) the adequacy of the tolerable intake defined (line 164) in hemodialysis patients, whose clearance is markedly lower?

110.  Page 6, line 164. Please spell out SCENIHR

No specific comments

Reviewer 3 Report

Considering the broad application of BPA and BPS in several medicinal products, including hemodialysis therapy, and the possible associated risks of these compounds especially in renal impaired patients, your study can clarify the concern of an unwanted leakage from dialyzers, but for a better understanding please explain the following aspects:

-     the statement regarding your research funding is missing.

-     better explain Table 2 – the meaning of precursor and product.

-     it is not clear what type of statistical analysis was used when comparing different types of dialyzers. In the article you only mentioned that “BPA/BPS were not detected in bloodline controls, cellulose-based membranes, and in Fresenius Medical Care’s FX CorAL 120 dialyzer” – what about the rest of the dialyzers?

-     in your article you presented the Margin of Safety and Tolerable Intake but it is not clear how these formulas were applied during your research. You explained the modality these measure determinations can be applied, but it is not clear if you performed these calculations.

-     even if in the beginning of your results you mentioned that 8 different dialyzers (from various manufacturers) were tested, your findings were focused especially on the newest dialyzer from Fresenius – what about the results assessed when analyzing the rest of the dialyzers?

Round 2

Reviewer 2 Report

The authors answered my questions to satisfaction

Reviewer 3 Report

Your manuscript was modified according to the suggested recommendations, and I thank you for all the clarifications.